# Nanoemulsion Loaded with Clotrimazole Based on Rapeseed Oil for Potential Vaginal Application—Development, Initial Assessment, and Pilot Release Studies

**DOI:** 10.3390/pharmaceutics15051437

**Published:** 2023-05-08

**Authors:** Michał Smoleński, Susanne Muschert, Dorota Haznar-Garbacz, Katarzyna Małolepsza-Jarmołowska

**Affiliations:** 1Department of Drug Form Technology, Faculty of Pharmacy, Wroclaw Medical University, Borowska 211A, 50-556 Wroclaw, Poland; 2Univ. Lille, Inserm, CHU Lille, U1008, F-59000 Lille, France

**Keywords:** nanoemulsion, clotrimazole, vaginal drug carriers, vaginal candidiasis, EVDF, vaginal drug delivery

## Abstract

Vaginal candidiasis (VC) is an emerging global hardly treated health issue affecting millions of women worldwide. In this study, the nanoemulsion consisting of clotrimazole (CLT), rapeseed oil, Pluronic F-68, Span 80, PEG 200, and lactic acid was prepared using high-speed and high-pressure homogenization. Yielded formulations were characterized by an average droplet size of 52–56 nm, homogenous size distribution by volume, and a polydispersity index (PDI) < 0.2. The osmolality of nanoemulsions (NEs) fulfilled the recommendations of the WHO advisory note. NEs were stable throughout 28 weeks of storage. The stationary and dynamic (USP apparatus IV) pilot study of the changes of free CLT over time for NEs, as well as market cream and CLT suspension as references, were conducted. Test results of the changes in the amount of free CLT released from the encapsulated form were not coherent; in the stationary method, NEs yielded up to 27% of the released CLT dose within 5 h, while in the USP apparatus IV method, NEs released up to 10% of the CLT dose. NEs are promising carriers for vaginal drug delivery in the treatment of VC; however, further development of the final dosage form and harmonized release or dissolution testing protocols are needed.

## 1. Introduction

Vaginal fungal infections (vaginitis) are emerging health issues in adult women, sexually active female adolescents, and young girls with comorbidities, e.g., type 1 diabetes or immune deficiency, as well as during antibiotic therapy [1,2,3]. Untreated or improperly treated vaginitis in pregnant women may also pose a serious risk to neonates due to invasive fungal infections that can be transmitted from mother to newborn [4]. Vaginal infections caused by the strains of *Candida* spp., mostly by *C. albicans* (more than eight out of ten cases), are called vaginal candidiasis (VC) [1,3]. A majority of women suffer from candidiasis at least once in their lifetime [5,6]. These infections are often recurrent and difficult to treat. Denning et al. and Foxman et al. highlighted the problem of recurrent vulvovaginal candidiasis (RVC), which is described as at least three or four following vaginitis caused by *C. albicans* per year. It is estimated that 15% of VC cases progress to RVC, resulting in 138 million women per year suffering from this disease [1,7,8]. Due to the limited and unclear information available on the treatment of RVC, extensive research is needed on new dosage forms, especially nanoforms, as well as drugs with higher antifungal activity and lower risks of side effects, e.g., ibrexafungerp and oteseconazole [9,10,11].

The most common and widely available drugs used in the treatment of VC are imidazole derivatives, such as fluconazole and clotrimazole (CLT) [12,13]. However, improper and extensive use of fluconazole has led to fungal resistance and therapeutic failures [1,13]. CLT is a weak base (pKa—4.7 and 6.0) with a molecular weight of 344.8 g/mol [14]. Despite its high antifungal activity, its applications are limited due to its poor water solubility (0.49 mg/L) and lipophilic properties (log P—6.1) [15,16]. The current recommendation for the treatment of uncomplicated VC with 1% CLT cream includes 5 g intravaginal daily application for 7 to 14 days [10]. Frej-Mądrzak et al. determined the susceptibility of the 125 isolates of *Candida* spp. to clotrimazole. Samples were obtained from the genitourinary tract of female patients and the minimal inhibitory concentrations (MIC) were measured. The MIC inhibiting the growth of 88% of yeasts was found to be <1 μg/mL, while MIC established for all tested isolates of *Candida* spp. was measured at 8 μg/mL [1].

The use of vaginal drug delivery, among its benefits such as rich vascularization bypassing hepatic circulation and a relatively high absorption area of 390 cm^2^, is a challenging route of drug administration [17,18]. Difficulties arise from small amounts (average 2 mL) of vaginal discharge typically present on vaginal mucosa, which strongly limits drug solubility [19]. Additionally, formulations are steadily washed out from the vaginal lumen by vaginal discharge due to its constant secretion (approx. 6 mL per day). It is necessary to design bioadhesive drug carriers able to deliver active pharmaceutical ingredients (APIs) in dissolved form to increase drug bioavailability and to prolong formulation residual time on the vaginal mucosa. Emulsion-based vaginal dosage forms (EVDF), an emerging and intensively developing research area, meet these criteria [20]. Different types of emulsions and emulgels are investigated as potential drug carriers for APIs [21,22,23,24,25,26]. Bachhav and Patravale [27], as well as Soriano-Ruiz et al. [23,28], have developed microemulsion-based gel, nanoemulsion (NE), and multiple-emulsion with CLT [23,28]. These formulations exhibited more favorable properties such as higher effectiveness against VC in vitro and in a small pilot study with patients (nanoemulsion), as well as a higher retention time in a vagina compared to market creams with CLT, i.e., Candid-V^®^ gel (Glenmark Pharmaceuticals Limited, Mumbai, India), Canesten^®^ (Bayer, Leverkusen, Germany), and Gine-canesten^®^ (Bayer, Leverkusen, Germany). However, there is still a strong need to continue to develop, improve, and investigate EVDF with CLT.

Most of the studies on EVDF are still in the initial laboratory phases and researchers are using unharmonized methods in the assessment of EVDF as European Pharmacopoeia (PhEur) does not provide detailed recommendations [20,29]. Therefore, the development of formulations and their evaluation methods is a parallel process. Among implemented dissolution/release/API availability testing protocols for NEs, dialysis bags are commonly used to determine the in vitro release/dissolution profile or simply the amount of available free API released from the encapsulated form in NE [30,31,32,33,34,35]. Based on a previous literature review, dialysis bags have been also used by several researchers in the evaluation of drug release in the case of local and systemic drug delivery via the vaginal route [20,23,36,37,38,39]. In the case of local drug delivery, dialysis bags might be considered questionable for the purpose of simulating biological membranes; however, as of now, there is no better alternative providing separation of free API from the encapsulated form in NEs’ oil droplets.

The research described in this paper is a matter of an international patent application (WO 2023/287309), and it has been recognized by the International Searching Authority as fulfilling the criteria of a novel, inventive step with industrial applicability [40]. NE and other types of emulsions for vaginal application are often converted into emulsion-based gels due to their low viscosity. The dosage form of emulgel is characterized by increased bioadhesion and vaginal residual time [26,36,41,42,43]. The aim of this study is to design and develop a nanoemulsion loaded with clotrimazole for vaginal application in VC, consisting of a safe and easily available lipophilic phase based on rapeseed oil as the basic, initial dosage form for further development towards the final clinical dosage form. Rapeseed oil has been chosen as a widely available natural source of omega-3,6,9 fatty acids, valuable phytosterols, and antioxidants [44]. Fatty acids are well known due to their anti-inflammatory properties and facilitation of tissue regeneration [45,46,47]. Moreover, rapeseed oil is stable at higher temperatures, unlike most vegetable oils; therefore, it is considered compatible with industrial manufacturing methods. The key factors affecting biopharmaceutical performance are the mean size of droplets (z-average), their uniformity, physiological pH, and osmolality; therefore, the optimal NE should have: (a) the droplet size < 90 nm to increase gravitational stability (phase separation due to coalescence and flocculation), (b) the absolute value of ζ-potential of 20 or higher to prevent droplets aggregation, (c) polydispersity index < 0.3 to provide sufficient uniformity, (d) physiological pH between 3.5 and 4.5, and (e) osmolality < 1200 mOsm/kg [48,49,50,51]. The droplet size lower than 100 nm will also prolong the residual time within the mucus mesh [52]. The first part of the paper includes an investigation of NE composition and the implementation of technological methods feasible to transfer to an industrial scale in the manufacturing of NE with CLT and the initial assessment of NE physicochemical properties. The second part of this research covers in-house designed pilot studies of the in vitro release of the free CLT from the encapsulated form in NEs’ oil droplets. Yielded NEs were compared with a market reference in terms of the cumulative release of API and the measured concentrations of free API with MIC of *Candida* spp.

## 2. Materials and Methods

### 2.1. Materials

CLT was purchased from Pol-Aura (Różanowo, Poland). Castor oil was supplied from Microfarm (Zabierzów, Poland). Rapeseed oil was purchased from Bunge Polska Sp. z o.o. (Kruszwica, Poland). In addition, 90% Lactic acid, Poloxamer 188 (Pluronic F-68) was purchased from Pol-Aura (Różanowo, Poland). Poloxamer 407 (Pluronic F-127), poly(ethylene glycol) 200 (PEG200), sorbitan monooleate (Span 80), and polyoxyethylenesorbitan monooleate (Tween 80) were purchased from Sigma-Aldrich (St. Louis, MO, USA or Steinheim, Germany). Propylene glycol was supplied by Firma Chempur (Piekary Śląskie, Poland).

Acetonitrile for HPLC (ACN) was purchased from S.Witko (Łódź, Poland). Acetone, methanol, and potassium phosphate monobasic (KH_2_PO_4_) were obtained from Chempur (Piekary Śląskie, Poland). Isopropanol was purchased from Pol-Aura (Różanowo, Poland). Milli-Q water (<0.05 µS/cm), used in batches manufacturing and all experiments, was self-produced from Hydrolab Ultra UV (Hydrolab Sp z o.o., Straszyn, Poland). All the chemicals and reagents used in this study were analytical grade.

Market reference, i.e., 1% CLT cream, MycoHydralin (Bayer Healthcare, Loos, France), was purchased in a local retail pharmacy in France.

### 2.2. Selection of the Formulation and Manufacturing Methods

The semiquantitative solubility of CLT in the different components was initially assessed. Approximately 1 g of the respective component was accurately weighed into the test vial, and 100 mg of CLT was added. The vial was vortexed for 2 min. The mixture was heated up to 60 °C and kept at this temperature for 15 min. Then, samples were again vortexed for 5 min and left for equilibration overnight. Approximately 100 mg of the mixture was accurately weighed into a 10 mL volumetric flask and dissolved in 6 mL of solvent made with acetone and methanol (40:60). The flask was filled up to the mark. The samples were analyzed with the HPLC method described in Section 2.10.

In order to screen for composition and manufacturing techniques providing a nanoemulsion, 22 formulations of different compositions and manufacturing methods were prepared. Rapeseed oil has been used as a lipid fraction of the potential nanoemulsions. Tween 80, Span 80, and Poloxamer F-127/F-68 were tested as potential surfactants and cosurfactants. Propylene glycol, PEG 200, and lactic acid were added to formulations as additional excipients. Within manufacturing techniques, high energy methods were used either alone or in combination with a low-energy method (initial mixing), or with another high energy method. The initial mixing of ingredients was performed with a magnetic stirrer (IKA Industrie und Kraftfahrzeugausrüstung GmbH, Königswinter, Germany). High-speed homogenization, HSH, (PRO250, PRO Scientific Inc., Oxford, CT, USA) and high-pressure homogenization, HPH, (GEA PandaPLUS 2000 lab homogenizer, GEA Mechanical Equipment Italia S.p.A, Parma, Italy) were used as high-energy methods. 

The compositions and manufacturing methods were designed using the general approach of the one-factor-at-a-time method.

### 2.3. Qualification of the Formulations for Further Studies

The qualification process was designed for initial fast screening and evaluation of optimal composition and manufacturing method(s). Visual appearance and stability in a centrifuge test were chosen as initial discriminatory criteria in the process of selection of the formulations.

Compositions of oil, surfactant, cosurfactant, demineralized water, and additional excipients were initially mixed using a magnetic stirrer, and then processed with (a) the high-energy method (high-pressure homogenization), (b) a combination of a low-energy (mixing with homogenizer PRO250 at 1500 rpm) and a high-energy method, or (c) a combination of both high-energy methods (HSH followed by HPH).

The visual appearance of a batch was assessed after equilibration, i.e., one day after the end of the manufacturing process—Criteria 1. Criteria 1 was used to reject macroemulsion and formulations exhibiting immediate phase separation. White and non-translucent batches were rejected. In the next step, the kinetic stability test of a nanoemulsion was performed—Criteria 2. The centrifuge test has been previously used by Wik et al. to initially assess the stability of NE by an acceleration of emulsion phase separation [53]. The test was implemented to reject formulations of low kinetic stability (metastable) by stress test, enhancing droplets aggregation and phase separation. Briefly, 1 mL of a formulation was placed in the Eppendorf tubes (*n* = 3). Samples were centrifuged for 30 min at 15,000 rpm in the thermostatic conditions of 25 °C using an Eppendorf Centrifuge 5417 R (Eppendorf AG, Hamburg, Germany). Batches with no signs of phase separation were qualified for further studies. 

### 2.4. Optimization of High-Pressure Homogenization Process

The high-pressure homogenization (HPH) process was evaluated in terms of CLT loss. The minimum pressure value required for NE formation of the qualified batch was found to be 1200 ± 100 bar. To establish maximum pressure values of the HPH process used for further reduction and providing homogeneity of NEs’ oil droplets, the NE premixes of S9:1-CLT, S8:2-CLT, and S7:3-CLT prepared according to Section 2.5 were homogenized using 3 different pressure levels—(a) 1300 ± 100 bar; (b) 1500 ± 100 bar; (c) 1700 ± 100 bar. Then, the obtained NEs were processed following the procedure described in Section 2.9 and the CLT concentration measurements using the HPLC method (Section 2.10) were performed. All measurements were taken in triplicate.

### 2.5. Manufacturing of Experimental S9:1, S8:2, and S7:3 Blank Formulations and Corresponding Batches with CLT for Further Assessment

The optimized blank NEs were prepared following the protocol in Figure 1. The premix of lipid fraction and cosurfactant was mixed and preheated to approximately 50 °C. Selected surfactant was added to the demineralized water and dissolved under magnetic stirring. Then, the aqueous phase and lipophilic phase were slowly mixed under magnetic stirring (approximately 600–1000 rpm). The obtained macroemulsion was left for at least 24 h at room temperature for equilibration. In the next step, the macroemulsion was homogenized using a high-speed homogenizer at 13,000 ± 200 rpm. The batch was cooled down to room temperature and, subsequently, high-pressure homogenization was performed. High-pressure homogenization was conducted in duplicate. The pressure was set to 1300 ± 100 bar. Batches with CLT were prepared using the same protocol (Figure 1), where the prescribed amount of the clotrimazole was accurately weighed and added to the lipid fraction premix and mixed on a magnetic stirrer for approximately 60 min.

### 2.6. Osmolality and pH-Level

Measurements of the osmolality of nanoemulsions with clotrimazole were performed using a Micro-osmometer Loeser TYP 6 (Loeser Messtechnik, Berlin, Germany). The batches were assessed 24 h after manufacturing. Samples were diluted two times with Milli-Q water before tests. All measurements were taken in triplicate.

The acidity of the batches was measured using a calibrated Mettler Toledo SevenMulti S40 (Mettler-Toledo LLC, Columbus, OH, USA) pH-meter equipped with a glass electrode. All measurements were performed at room temperature and conducted by direct immersion of the electrode in the glass bottles with NEs until a stable result was yielded. All measurements were taken in triplicate. A range of pH of 3.5 to 4.5 was considered physiological.

### 2.7. DLS Droplet Size Determination and Zeta Potential Measurements

The average droplet size (z-average), droplet size distribution by volume, polydispersity index (PDI), and zeta potential (ζ-potential) of the obtained nanoemulsions were measured at 25 ± 0.1 °C by Dynamic Light Scattering (droplet size and PDI) and Electrophoretic Light Scattering (ζ-potential) methods using Zetasizer Nano-ZS ZEN3600 (Malvern Instruments Ltd., Worcestershire, UK). The Non-Invasive Back Scatter method, i.e., 173 degrees laser configuration, was used. Samples were diluted 50 times with Milli-Q water to fit the instrument’s sensitivity range and measured within 30 min after dilution to prevent the possible agglomeration or precipitation of nanoemulsion components. All measurements were taken in triplicate.

### 2.8. Stability Test

The stability test of batches was carried out in the conditions of 25 ± 2 °C and relative humidity of 30–65%, protected from light for 28 weeks. Then, 200 ± 5 g of each NE was placed in 250 mL borosilicate glass 3.3 lab bottles with high-temperature-resistant screw caps and sealed using parafilm to avoid the influence of humidity on the stability. Visual appearance (to screen for macroscopic signs of phase separation, precipitation of components, etc.), z-average, size distribution by volume, PDI, ζ-potential, and pH level were evaluated. The following parameters were assessed in 3-time points: (a) 24 h after manufacturing (0-time), (b) after 4 weeks, and (c) after 28 weeks. All measurements were taken in triplicate.

### 2.9. Sample Treatment in CLT Assay

Approximately 1000 mg of NE was exactly weighed into a 25 mL volumetric flask. The solvent (1:1 mixture of isopropanol and acetone) was added to the flask and mixed until the NE was fully dissolved. The flask was filled up to the mark with the solvent. Then, 1 mL of the obtained solution was transferred to the 5 mL volumetric flask and filled up to the mark with acetonitrile (ACN). Samples were filtered with 25 mm, 0.22 μm, syringe filters made of regenerated cellulose. 

### 2.10. HPLC Method for CLT Determination

The concentration of CLT in various media was determined by the reverse phase HPLC method. The analysis was performed using a Shimadzu LC-2050C (Shimadzu U.S.A Manufacturing Inc., Canby, OR, USA) equipped with a DAD detector and Phenomenex Gemini C18, 150 mm × 4.6 mm, 3 μm column. Elution was isocratic and the mobile phase consisted of ACN and 20 mM phosphate buffer pH 6.8 (60:40), the flow rate was 1.5 mL/min. The injection volume was 20 μL. CLT detection was recorded at a retention time of approximately 5 min at 265 nm. The method was validated within concentrations of 2–300 μg/mL with a linearity of R^2^ = 0.9999 in terms of accuracy, precision, repeatability, specificity, and linearity according to EMA and ICH guidelines (ICH Topic Q 2 (R1) Validation of Analytical Procedures: Text and Methodology, Note CPMP/ICH/381/95 for Assay).

### 2.11. The Pilot Study of the Changes in the Amount of Free CLT Released from Encapsulated Form—Stationary Method

The study was designed in-house and performed with batches S9:1-CLT, S8:2-CLT, S7:3-CLT, and references—1% CLT cream (MycoHydralin, Bayer Healthcare, Loos, France) and 1% suspension of API powder with MilliQ water as the vehicle. Servapor dialysis bags made of regenerated cellulose with a 10,000–12,000 Daltons cut-off and 2.5 nm pores (Serva Electrophoresis GmbH, Heidelberg, Germany) were used to separate the free CLT from the encapsulated form. The test setup consisted of cylinder-shaped vessels placed in a heating bath and a magnetic stirrer under the bath. Then, 300 mL of 50 mM phosphate buffer pH 3.5 was used as a dissolution medium. The bath temperature was set to 37 °C and the stirring speed was set to 100 rpm. The pilot study test was conducted for 5 h to simulate a short contact time of formulation with the vaginal mucosa. The following study protocol was developed: 7-cm long dialysis bags were conditioned for 1 h in the medium prior to the test; 5 mL of the formulation was put into the dialysis bag and the membrane was sealed with clips. Then, the bags were transferred to the vessels. Samples were withdrawn every 30 min of the test. The single sample volume was set to 5 mL. Samples exhibiting disruption of membrane integrity were rejected. The same volume of fresh medium was added to the vessels after collecting every sample. Samples were diluted 1:1 with ACN and filtered with 25 mm, 0.22 μm, syringe filters made of regenerated cellulose. The concentration of CLT was analyzed using the HPLC method described in Section 2.10. The test was conducted in triplicate for each formulation.

### 2.12. The Pilot Study of the Changes in the Amount of Free CLT Released from Encapsulated Form—Dynamic Method with USP IV

Additional tests of all batches and references—1% CLT cream (MycoHydralin, Bayer Healthcare, Loos, France) and 1% suspension of API powder were performed using the USP apparatus IV setup; i.e., flow-through cell to simulate dynamic conditions. The setup consisted of flow-through cells (22.6 mL cell, CE 1, Sotax AG, Basel, Switzerland) conditioned at 37 °C by means of a water bath and a piston pump (CY 7-50, Sotax AG) to provide a stable flow rate of 2 mL/min. The test was conducted for 5 h. We used an open circuit to provide fresh 50 mM phosphate buffer pH 3.5 to the samples to increase CLT solubility and mimic the wash out conditions. Then, 2 mL of the NE formulation was put in the dialysis bag, as described in Section 2.11. The scheme of the apparatus is presented in Figure 2. Each batch and reference were tested in triplicate. The samples were collected every 10 min in the first hour of the experiment and, later, every 30 min up to 5 h. Samples were processed and analyzed as described for the stationary method.

### 2.13. Statistical Analysis

All experiments were conducted in triplicate. The mean values and standard deviations were calculated for the results of all conducted experiments. The ANOVA test was carried out at *p* ≤ 0.05 for the results of pilot studies of the changes in the amount of free CLT released from encapsulated form. The analysis was performed using the built-in tools of Microsoft Excel software.

## 3. Results

### 3.1. Process of Developing the Formulation Composition, Manufacturing Method, and Selection of Obtained Nanoemulsions for Further Studies

The results of the semiquantitative determination of CLT solubility in tested components are presented in Table 1.

The initial assumption was that the amount of oil in the formulation should provide sufficient solubility of CLT (10 mg of API per 1 g of formulation). Therefore, the minimum concentration of rapeseed oil was set to 20% (*w*/*w*) of the formulation. The addition of CLT solubilizer (PEG 200 or propylene glycol) has been considered at different levels from 0.5% (*w*/*w*) for PEG 200/propylene glycol and 6.6%, 10%, and 20% for PEG 200 only. The sets of formulations are presented in Table 2, Table 3, Table 4, Table 5, Table 6, Table 7, Table 8 and Table 9. All compositions were balanced with MilliQ water up to 100% (*w*/*w*).

The first set of formulations is presented in Table 2. The high molecular weight poloxamer (Pluronic F-127) was used as a surfactant due to its low irritating properties, vast industrial applicability, and bioadhesive properties. HPH was used as the first manufacturing high-energy method at three levels: 400 ± 100 bar, 800 ± 100 bar, and 1200 ± bar.

As the obtained formulations failed to meet criteria 1, the second set of formulations was designed (Table 3). The high-energy method (HSH) was preceded by initial mixing at a low speed of 1500 rpm using a homogenizer PRO 250. The amount of oil was reduced to assess the applicability of the methods.

Due to the failure to comply with criteria 1, the third set was designed, where two high-energy methods, i.e., HSH and HPH, were used. The results are presented in Table 4. PEG 200 was included as a CLT solubilizer to investigate the influence of API addition on the formulation.

Despite two high-energy manufacturing methods, formulations 8 and 9 failed to meet criteria 1. We concluded that the use of a single surfactant was the reason for the failure. The fourth set (Table 5) was prepared. The fourth row of formulations was characterized by the addition of Tween 80 as the second surfactant in a 1:1 mass ratio and the single high-energy method (HPH).

As all formulations among the fourth set failed to comply with criteria 1. We concluded that the cosurfactant of a low HLB number might be more favorable; therefore, we replaced Tween 80 with Span 80. The combination of two high-energy methods was also implemented to provide more kinetic energy into the system. The fifth set (Table 6) was designed. The concentration of rapeseed oil was tested at two levels—20 and 40% (*w*/*w*).

As the results were unsatisfactory, we decided to increase the total amount of surfactant mix (see Table 7).

Formulation 16 was able to meet criteria 1, but failed to meet criteria 2 (centrifuge stability test). This suggested that the amount of rapeseed oil in the system was too high. We decided to reduce the concentration to the lowest, i.e., 20% (*w*/*w*), and prepared the seventh set of formulations—Table 8.

We obtained NE, fulfilling criteria 2 yet characterized by an inhomogeneous structure due to the gelling of Pluronic F-127 as the result of high temperature during HPH. To resolve that issue, we replaced Pluronic F-127 with one of lower molecular weight—Pluronic F-68 (Table 9).

The compositions of selected blank nanoemulsions were further optimized. The final composition of the selected batches is presented in Table 10 and the corresponding batches loaded with clotrimazole are summarized in Table 11. The selected nanoemulsions differed in terms of their surfactant to cosurfactant ratio, i.e., 9:1 (batch S9:1), 8:2 (batch S8:2), and 7:3 (batch S7:3).

The results of the drug loading efficiency depending on different levels of pressure during HPH (following procedures 2.5 and 2.9) are presented in Table 12. The results showed a significant impact on CLT loading in the NEs. The rising of the pressure level over 1300 bar during the HPH resulted in the loss of the drug. Therefore, the manufacturing process was adjusted as described in Section 2.5.

### 3.2. Physico-Chemical Evaluation of the Batches

All yielded batches were translucent and opalescent (see Figure 3), suggesting favorable properties of the small size of NEs’ droplets [48,54,55,56]. The exact values of the physico-chemical parameters of the obtained batches right after manufacturing are presented in Table 13. All formulations were characterized by relatively small droplet diameters within the range of 100 nm, confirming visual observations. The obtained blank formulations were homogenous and the recorded PDI was within a range of 0.089 to 0.147 with narrow standard deviations. The results of a more in-depth investigation of droplet diameters and uniformity are presented in Figure 4A. Batch S8:2 had the smallest mean droplet diameter, as well as the narrowest size distribution by volume among blank NEs. The largest droplets were ≤100 nm, while most of the droplets were ≤50 nm. S9:1 and S7:3 were characterized by similar and wider droplet size distributions by volume than S8:2. The largest droplets were ≤175 nm, while most of the droplets were ≤100 nm. Higher differences between blank NEs and NEs with CLT were observed for ζ-potential levels. Batches S9:1 and S7:3 had low negative values of ζ-potential close to the neutral, contrary to S8:2, where a positive value of approximately 16.5 mV was recorded. The acidity level of blank NEs was within the physiological range.

The addition of CLT had a significant impact on the z-average, the distribution of droplet size by volume, ζ-potential levels, and the pH level. In all three formulations, mean droplet diameters decreased by approximately 5–22%. The most significant difference was observed for the NE with a 9:1 surfactant-to-cosurfactant ratio. The distribution of droplet size in all three batches with CLT was unified and almost identical (Figure 4B). The largest recorded droplets were ≤150 nm, while most of the droplets were ≤75 nm. The addition of API to the formulations slightly elevated PDIs in batches S9:1 and S8:2; the most significant change was observed in batch S7:3, where PDI increased by approximately 50%. It had also noticeably increased the levels of ζ-potential, changing the values of approximately +22 mV, +20 mV, and +23 mV in the case of formulations S9:1-CLT, S8:2-CLT, and S7:3-CLT, respectively. The change in ζ-potential corresponded with higher pH of approximately 4.0–4.2 compared to blank NEs. In general, the acidity of formulations was influenced by the surfactant-to-cosurfactant ratio. The pH decreased with the increasing content of Span 80.

The osmolality was measured for NEs with CLT. All batches were characterized by osmolality within the range of 750–880 mOsm/kg. The highest value of osmolality was recorded for batch S9:1-CLT, while the lowest was for S7:3-CLT.

### 3.3. Results of the Stability Test

The change in the physico-chemical properties of tested batches is summarized in Table 14. The visual appearance of the NEs with CLT did not markedly change and no macroscopic signs of precipitation or phase separation were recorded.

The mean droplet diameter had no tendency to increase or decrease within 28 weeks. PDI values of batches S8:2-CLT and S7:3-CLT dropped. However, those changes had no visible impact on size distribution by volume over time (Figure 5), which remained constant for all tested batches. ζ-potential showed greater variability over time. After one month, ζ-potential decreased by 0.7 mV (S7:3-CLT) to 4.6 mV (S9:1-CLT), which was not related to changes in the acidity, as pH values did not significantly shift during the same period. ζ -potential increased close to initial values after 28 weeks in the case of S9:1-CLT and S7:3-CLT. For the batch S8:2-CLT, an increase in ζ-potential of 4.5 mV from the initial value was observed, reaching 24.20 mV. Although the S7:3-CLT had the smallest variation in values over the 28 weeks of observation, there was a high standard deviation at the last time point. During the measurement of ζ-potential, two fractions of the droplets with positive and negative surface charges were recorded. A slight change over time in the pH levels was observed in the S7:3-CLT batch.

### 3.4. Pilot Study of the Changes in the Amount of Free CLT Released from Encapsulated Form—Stationary Method

The results of the pilot study of the changes in the amount of free CLT over time are presented in Figure 6. The release kinetics are similar for all tested NE batches (*p* = 0.51). All NEs yielded a mean maximum amount of released nonencapsulated API of approximately 25% in 300 min. Despite similarities, some tendencies were observed. The mean maximum amount of released API for batch S9:1-CLT was slightly lower when compared to other NEs. A high standard deviation for each NE was noticed, the highest was observed for batch S8:2-CLT, while the lowest standard deviation was recorded for batch S9:1-CLT. NE, with a 9:1 surfactant-to-cosurfactant ratio, started to release the API through the membrane in approximately 60 min, while in the case of the other batches, CLT in the medium was recorded within 30 min. Additionally, 1% suspension of CLT powder used as a reference presented similar kinetics to NEs-CLT batches (*p* = 0.67), in contrast to cream with 1% CLT, which released only 5% API of the total dose in 5 h. The cream was significantly different from NE batches and 1% CLT suspension (*p* < 0.05).

### 3.5. Pilot Study of the Changes in the Amount of Free CLT Released from Encapsulated Form—USP IV Method

The results of the pilot study conducted with the USP IV setup are presented in Figure 7. The profiles of changes in the free CLT concentration of tested batches are similar and characterized by the incomplete release of the API (*p* = 0.35). The standard deviations were low in all tested formulations, except for batch S9:1-CLT, which also reached the highest level of released CLT, up to 10% of the dose; however, those differences were statistically insignificant. The performance of batches S8:2, S7:3, and 1% suspension were similar and yielded slightly lower levels compared to the S9:1-CLT (*p* = 0.74) and significantly higher than 2–3% of the released dose of API in the case of cream with CLT (*p* < 0.05). We recorded the almost immediate onset of the release of the CLT for all batches.

### 3.6. Comparison of Changes in Free API Concentration over Time with MIC

The measured concentrations of nonencapsulated CLT are presented in Figure 8. In the stationary method, significantly higher levels were recorded. S8:2-CLT and S7:3-CLT reached a concentration higher than 1 μg/mL in 30 min, while S9:1-CLT and CLT reached suspension in 60 min. Market reference required a significantly longer time (*p* < 0.05); i.e., 120 min to achieve that level. Concentrations higher than 8 μg/mL were reached by (a) S8:2-CLT within 30 min; (b) S7:3-CLT and 1% CLT suspension within 60 min; (c) S9:1-CLT within 90 min; and (d) MycoHydralin cream within 120 min.

The concentrations measured in flow-through experiments were significantly lower than those observed in the stationary method. The level of MIC CLT 88% was recorded within 10 min of the experiment in the case of all tested samples. None of the tested formulations were able to reach the concentration of 8 μg/mL. A similar and almost constant level of 3–4 μg/mL was observed for NEs batches and 1% CLT suspension. The cream with CLT reached a significantly lower (*p* < 0.05) and constant concentration of 1 μg/mL. It is worth noting that this method was performed under a constant renewal of the dissolution medium.

## 4. Discussion

According to the established testing protocol (Section 2.3), the most promising composition consisted of rapeseed oil and Span 80 as the lipophilic phase and an aqueous solution of Pluronic F-68 as the hydrophilic phase of nanoemulsion. The addition of lactic acid and PEG 200 to the formulation improved the physiological parameters of the formulation and the cross-mixing of components, respectively. HSH and HPH turned out to be a suitable combination of high energy methods in the manufacturing of NEs with CLT. However, the implementation of HPH requires confirmation of drug loading due to possible loss of some API [57]. The acceptable deviation was assumed at ±5%. In this work, the maximum pressure value not influencing the drug loading was found to be 1300 bars, where 97% of the initial dose was observed. A further increase in pressure level resulted in a significant loss of the API to approximately 50% of initial drug loading. The exact mechanism of the CLT loss was not investigated by us. Possibly, when a formulation is pushed through a microns-sized homogenization gap under high pressure, the batch is subjected to extreme kinetic forces resulting from acceleration to a very high velocity. This stress causes disruption of the larger oil droplets into the nano-sized ones. A part of the energy is also transferred into heat. The higher the pressure applied, the more disruptive forces act on the formulation, which might result in unfavorable processes, such as the aggregation of droplets or other particles, coalescence, water evaporation, and other interactions [57,58,59]. Pressure exceeding a critical level might lead to precipitation of the API and, consequently, the CLT could be filtered out by the narrow homogenization gap and/or interaction with the tubing system. However, this needs to be confirmed by the appropriate experiments.

The yielded blank and drug-loaded NEs were acceptably homogenous based on low PDI values and small standard deviations [60]. The mean droplet diameter of blank and loaded NEs was smaller than 100 nm. Particles with a size < 100 nm are expected to enter and slowly penetrate vaginal mucus. Larger droplets (200–500 nm) would penetrate to vaginal mucosa more rapidly, and hence, increase absorption of the CLT to the circulatory system, which, in this case, is not considered beneficial due to the possible systemic side effects [18]. Entrapment of CLT in the mucus mesh could increase its residual time and, consequently, enhance the antifungal therapy of vaginitis. Within blank NEs, the most promising properties were observed in the case of batch S8:2, as its z-average was the lowest and the most homogenous based on droplet size distribution by volume. The ζ-potential of blank batches S9:1 and S7:3 indicates their possible instability, as the values were less than ±10 mV and close to the point of zero charges [49]. Batch S8:2 yielded the highest ζ-potential of approximately +16.50 mV, indicating its relative stability and the most favorable composition among the blank batches. The addition of CLT had a strong influence on the NEs properties, as it affected the z-average and ζ-potential, narrowed the size distribution by the volume of batches, and shifted the distribution peaks towards smaller diameters (except S8:2). The unique properties and differences in terms of the z-average, size distribution by volume, PDI, and ζ-potential of batch S8:2 might be a result of the surfactant-to-cosurfactant ratio. Zeng et al. described that the surfactant-to-cosurfactant ratio might contribute to a decrease in the interfacial tension and an increase in surfactant layer flexibility, which results in a system of higher entropy [61]. After the addition of the API, the shape and values of the droplet size distribution by volume of the tested batches became almost identical (see Figure 4). In our opinion, the increased absolute value of ζ-potential, together with changes in mean droplet size, suggest that CLT has stabilizing properties on these NEs compositions. In nanoemulsion loaded with clotrimazole obtained by Soriano-Ruiz et al., the addition of CLT also affected the size of the droplets by increasing their diameter [23]. Borhade et al. studied the influence of the CLT, the composition of NEs, and the pH of the media on the properties of NEs. Authors have discovered the significant impact of CLT loading on NEs. In some cases, especially in media at a pH of 1.2, CLT decreased the mean size of the droplet of NE. This phenomenon was explained by the protonated imidazole group of CLT in the acidic media, which resulted in increased CLT solubility [14]. Although clotrimazole increased the absolute value of the ζ-potential of NEs, values lower than ±30 mV may not guarantee long-term stability, yet ranges ±10 mV to ±20 mV and ±20 mV to ±30 mV may be considered relatively stable and moderately stable [49]. The API, as a weak base, has also elevated the pH of the formulations to mid-values (approximately 4.0) of the physiological range of 3.8 to 4.5 [50]. Change in acidity is one of the major factors affecting ζ-potential, and therefore, the observed changes in the ζ-potential values may result from the CLT ionized form in acidic pH [14]. This observation may also suggest that a part of the API dose is localized close to the lipophilic phase boundary.

Evaluation of the osmolality of a vaginal formulation is an important factor, as hyperosmolal formulations have been reported to have irritating properties on vaginal mucosa, and therefore, could cause unacceptable side effects [62]. To the best of our knowledge, this is the first study which investigated the osmolality of EVDF. Developed batches meet the World Health Organization advisory note on the “Use and procurement of additional lubricants for male and female condoms: WHO/UNFPA/FHI360”, as measured osmolality was lower than 1200 mOsm/kg, yet higher than the ideal < 380 mOsm/kg [51]. When considering osmolality levels, the irritation potential on membranes decreases in a series of S9:1-CLT, S8:2-CLT, and S7:3-CLT from approximately 870 mOsm/kg to approx. 760 mOsm/kg. PEG 200, Pluronic F-68, and lactic acid, and possibly CLT, have been identified as significant osmotic agents [63,64,65,66]. The concentrations of PEG 200, lactic acid, and CLT were constant for NEs. Only the levels of Pluronic F-68 varied between tested batches. Osmolality values corresponded with the Pluronic F-68 loading as the highest osmolality (871 ± 25 mOsm/kg) was recorded for S9:1-CLT, which contained 18% *w*/*w* of poloxamer, while the lowest osmolality (758 ± 5 mOsm/kg) was measured for batch S7:3-CLT, containing 14% *w*/*w* of the excipient. Despite the osmolality of NEs fulfilling the recommendations of the WHO, the potential irritating properties of components were investigated. Surfactants in high concentrations have been recognized as potentially irritating or even toxic substances [67]. The implementation of Pluronic F-68 in high content (>20%) in a vaginal formulation containing nanoparticles was previously investigated by Zhang et al. [68], where no irritating properties of this excipient were observed in vivo on rabbits. The safety of Span-80 was evaluated by Minamisakamoto et al. [69] in the study of noisome, where insignificant toxicity of a high (46.5%) concentration of the surfactant was observed. This indicates NEs should not exhibit irritating properties on vaginal mucosa, as the concentration of surfactants used in our study is lower than reported in previous research.

The evaluation of the stability of NE is one of the most important formulation properties to evaluate, as nanoemulsion will eventually separate into oil and aqueous phases. In ideal NE, this phase separation process is slow enough to not have an impact on the formulation performance in the pharmaceutical industry. The main processes involved are flocculation, coalescence, and Ostwald ripening [48,54,55,56,70]. In general, these processes lead to aggregation or combining of the droplets, and consequently, creaming and phase separation [48,54,55,56,70]. Increasing the mean droplet size (z-average), PDI, and size distribution could be one of the markers indicating NE disruption. Changes in ζ-potential values (especially a drop in absolute value or the appearance of opposite charges) associated with variations in pH levels might accelerate or initiate emulsion destabilization. During the stability studies, the variability over time in terms of visual appearance, z-average, PDI, mean droplet size distribution by volume, ζ-potential, and pH-value was assessed to monitor early signs of NE destabilization. Within the first month, batches exhibited acceptable stability and no significant changes were observed in the evaluated properties and parameters, except for ζ-potential. The decrease by approximately 21%, 7%, and 3% of ζ-potential values, compared to their initial values of batches S9:1-CLT, S8:2-CLT and S7:3-CLT, respectively, indicates that destabilizing processes might have occurred in the batches. Variations in ζ-potential values did not correspond with constant pH levels. However, no signs of API precipitation, phase separation, or flocculation were observed. After 28 weeks, an increase in ζ-potential values was observed in all three NEs with CLT. The previous drop in ζ-potential levels might have been related to the equilibration of the formulations. Higher values of ζ-potential compared to the initial levels of batch S8:2-CLT with no change in the pH suggest that this composition is more favorable when compared to 9:1 and 7:3 surfactant and cosurfactant ratios. Although the values of ζ-potential in the case of S9:1-CLT returned to around initial values, a higher standard deviation was observed. The increase in acidity was at the limit of acceptable variability of ±0.05. In the case of S7:3-CLT, an increase by 0.6 mV compared to initial values of ζ-potential was recorded after 28 weeks. Despite the small change in value, an increase in the standard deviation was observed resulting from the separation of the surface charge peaks into fractions with positive and negative charges. These changes were accompanied by a decrease in pH of 0.08. These together indicate the onset of destabilizing processes and the inability of the 7:3 surfactant-to-cosurfactant ratio to provide stability for NEs in the long term. Analysis of the droplet size distributions over time revealed no significant changes for all NEs with CLT, which may be related to the stabilizing phenomena of the API.

Published studies regarding emulsion-based vaginal dosage forms (EVDF) are highly variable [20]. The volume of approximately 2 mL of vaginal discharge present in the vaginal lumen is 250–500 times less than the volume in commonly used apparatus such as USP I and USP II. Several researchers have tried to reduce this difference by designing a new simple setup consisting of dialysis bags with EVDF inside and a vessel/container with dissolution media [36,71]. In our pilot study, the dialysis bag was used to separate free CLT in medium from encapsulated form, but not to simulate systemic absorption. Another issue with a simulation of physiological conditions is related to the constant wash-out of the formulation from the vaginal lumen. This could be simulated by replacing the withdrawn sample volume with fresh media. In the case of pilot studies, we chose a 50 mM phosphate buffer as simulated vaginal fluid. The pH was adjusted to 3.5 to increase CLT solubility (based on our previous R&D studies). Due to the poor solubility of clotrimazole in aqueous media, we had to increase the volume of the buffer to 300 mL to avoid a possible limitation of the study due the risk of the saturated medium. The study was designed using standardized apparatus and equipment to provide methods transferable to industrial protocols and to enable a focus on the technological aspects of NEs. No significant differences were observed between the batches (*p* > 0.05), which showed high standard deviations. The high impact of the membrane of the dialysis bag was one of the major factors resulting in variability in drug release between the three samples analyzed. The selection of a dialysis membrane for the pilot studies was based on the formulation properties, and the purpose was not to simulate vaginal mucosa. Yet, this membrane was chosen due to the low sorption of CLT to regenerated cellulose and a small pore diameter of 2.5 nm, preventing an uncontrolled passage of NEs to the phosphate buffer. However, the obtained NEs revealed significant osmolality resulting in osmotic pressure between the NEs inside the dialysis bags and phosphate buffer outside, which created an influx of dissolution medium to the formulation. Changes in osmotic pressure have been described in several reports on colloids or nanomaterials encapsulated in dialysis bags [72,73,74]. In fact, we observed a swelling of the dialysis bags exclusively in samples containing NEs and did not record any swelling in the case of CLT suspension and cream (data not shown). Osmotic pressure may have caused micro-damages in the structure of the membrane and resulted in high standard deviations. Comparatively high SDs were not observed for the suspension and cream. Therefore, further investigation is necessary to avoid the influence of osmotic pressure on dialysis membrane integrity. Despite this fact, we were able to observe some insignificant differences in the release curves. The highest level of released API was observed in batch S7:3-CLT and S8:2-CLT (about 27% of CLT dose). Suspension of the CLT reached a similar level of released API to the S9:1-CLT batch (approximately 25% of the initial CLT loading). However, this difference is considered statistically insignificant (*p* > 0.05), which is also visualized in overlapping standard deviations. Cream with CLT presented a significantly lower performance (*p* < 0.05) in the pilot stationary test and released only approximately 5% of the initial dose of CLT. It is obvious that suspension is not considered a potential intravaginal formulation due to the almost immediate wash out from the vaginal lumen, the poor solubility caused by limited vaginal discharge, and possible irritating properties. The market reference—cream with CLT—yielded a limited release of API, possibly due to the poor and slow diffusion of API through the formulation and the reduced penetration of the medium/vaginal discharge into the cream.

The stationary method was not discriminatory to distinguish the release kinetics of free CLT from the encapsulated form of the different batches. An additional test with a USP apparatus IV setup was performed to evaluate the resistance of the NEs to the wash out by vaginal discharge, the sensibility to stress conditions, and the role of the hydrophilic/lipophilic surfactants ratio in solubility kinetics and membrane permeability. The constant flow of medium in an open circuit simulates the potential of the formulations to confirm their activity, i.e., the constant availability of API. Surprisingly, despite the higher total volume of the medium (≈600 mL per single sample), the levels of free API transferred to the medium were significantly lower by approximately 2.5 times compared to the stationary method. The statistical results for the NE batches and 1% CLT suspension obtained using the stationary and flow-through methods are significantly different (*p* < 0.05), whereas, in the case of cream with CLT, the results are similar (*p* = 0.10). However, the dynamic release method provided more consistent profiles due to the low standard deviations. The small variations are explainable by the fact that the latter were placed inside the flow-through cell which, due to its small volume of 18 mL, prevents the membrane from expanding. In the flow-through method, the most favorable performance was exhibited by the S9:1-CLT batch, while the batches S8:2-CLT and S7:3-CLT yielded similar and slightly lower concentrations in contrast to the stationary method, where S9:1-CLT presented the lowest mean values of released API among NEs, and S7:3-CLT had the highest concentration. This phenomenon is worth further investigation. One of our hypotheses involves the role of the compositions of the surfactants. CLT is more soluble in Span 80 than in poloxamers, and Span 80 as a hydrophobic surfactant is less miscible with water. The constant flow of fresh medium reduces the concentration of Span 80 in the medium on the outer side of the membrane, reducing the ratio of CLT penetration through the membrane of the dialysis bag. Pluronic F-68, as a hydrophilic surfactant, can diffuse more rapidly to the phosphate buffer medium and provide enhanced solubility compared to the pure phosphate buffer. The highest concentrations reached by S9:1-CLT might support this thesis. However, Pluronic F-68 is not able to solubilize CLT as much as Span 80, which resulted in lower levels of free API despite the overall larger volume of the acceptor medium. In the stationary method, where the medium was not entirely replaced with fresh buffer, the higher concentration of the lipophilic surfactant provided higher solubility of CLT, resulting in the batch S7:3 reaching the highest concentrations. However, this hypothesis requires further investigation in more discriminatory conditions to confirm the strong influence of surfactant composition on the differences in free API release kinetics. Another factor that may have had an impact is osmotic pressure, yet in both pilot tests, comparable swelling of the dialysis bag was observed. CLT cream reached the lowest and similar concentrations in both tests. This confirms that NEs are drug carriers feasible to provide higher doses of lipophilic substances compared to vastly used creams in the treatment of vaginal candidiasis.

The results of the pilot release studies were also presented in the form of CLT concentration in the medium of phosphate buffer. The purpose was to compare the free API levels obtained in the tested formulation depending on the release method used. Bearing in mind that the tests were not designed to simulate physiological conditions, the aim of these studies is to assess the kinetics of the release of free CLT from encapsulated form. Additionally, the concentrations were compared with MIC of *Candida* spp. for CLT. The increasing concentration among the experiments in the stationary method proves that the amount of medium was providing sufficient solubility for the CLT. NEs provide higher availability of the nonencapsulated CLT form when compared to cream. This short method was tested as a potential replacement for the time-consuming diffusion agar plate tests when the MIC was previously established. The MIC values were only used as the reference points. An additional benefit of this method is the lack of analytical difficulties related to the extraction of the API from the agar during sample preparation, e.g., HPLC measurements or the selection of the sampling point on the agar plate. In the stationary method, we recorded the free CLT concentration (>8 μg/mL) effective against all isolated yeast strains by Frej-Mądrzak et al. [1] after 30–90 min in the case of NEs and CLT suspension, or at the end of the experiment in case of MycoHydralin cream. Although the antifungal level of CLT for the majority of strains (1 μg/mL) was reached within 1 h of the test for all formulations, except the market reference, less favorable observations were recorded in the dynamic pilot study. The highest measured concentration of CLT did not reach the value of 8 μg/mL. However, using the flow-through method, we observed the rapid release of free CLT within 10 min, in the amount providing MIC against most of the strains. In the stationary method, it must be taken into consideration that the total volume of the medium was 150 times greater than that found in physiological conditions, therefore potential concentrations in physiological conditions could be higher. However, this pilot release test was designed to assess the technological properties of NEs, such as release kinetics considering limited CLT solubility. Consequently, the excess of the medium was reduced from 500 times greater than physiological to 150 times. In the USP apparatus IV setup, the effective volume of phosphate buffer inside the flow-through cell (capacity of the cell minus the glass beads) was approximately 18 mL, which was only six times higher than physiological, but the highest concentration of CLT was only 4 μg/mL. This might be due to the high flow of the medium; i.e., 2 mL/min resulting from the pump properties. However, this experiment confirmed that part of the dose of free CLT is instantly available in contrast to the cream-based formulation. When considering future tests simulating physiological conditions, it would be beneficial to prepare the setup providing a slower flow of the medium inside the cells to simulate physiological conditions of the vagina more accurately, yet the total volume of the medium should enable sufficiently frequent sampling. 

In summary, the obtained NEs met the initial requirements resulting from key factors of biopharmaceutical performance, i.e., a z-average of 50–60 nm, a narrow size distribution (high uniformity of the droplet size), a physiological pH of approximately 4.0, and osmolality within the range of 700–900 mOsm/kg. The most important factors characterizing the NEs performance within the study of the release of free CLT from the encapsulated form are the fast onset of CLT release from NE and the ability to reach MIC or a higher CLT concentration. Although all of the NEs-CLT batches presented similar performances, the formulation S8:2-CLT seemed the most promising, as it exhibited slightly lower variability of ζ-potential between blank and drug-loaded formulations, the highest ζ-potential and, unlike S7:3-CLT, did not show peaks of a negative charge after 28 weeks of stability evaluation. S8:2-CLT showed a fast onset of the release of the free CLT and reached the concentration > 8 μg/mL of CLT considerably earlier than other batches in the stationary method.

Another problem which has been noticed in our study is the lack of recommended methods for evaluation of the release of a free drug form from vaginal drug carriers. We used two methods of dissimilar dynamic and medium flow. We observed inconsistent results and none of the methods were discriminatory for tested batches. These methods together and alone could not simulate physiological conditions; i.e., the volume and secretion rate of vaginal discharge. It is important to develop a method that more accurately simulates the vaginal environment, i.e., the low volume of the ambient medium in the test cell, yet provides a sufficient amount for the collection of samples and a medium flow simulating the physiological exchange of vaginal discharge (6 mL/24 h) [75,76,77].

## 5. Conclusions

This study investigated the technological aspects of NEs with CLT. Tested batches were manufactured using methods feasible to transfer to an industrial scale. Stable NEs with favorable properties (mean droplet size, homogeneity, physiological pH, and osmolality) were obtained. Although the obtained NEs were similar, the most promising properties revealed batch S8:2-CLT as being suitable for future development. NEs are emerging potential drug carriers in the treatment of local vaginal conditions. There are no recommendations for API release or the dissolution testing of EVDF. Stationary and flow-through methods provided incoherent results. It is crucial to develop a robust drug release testing protocol which will provide more discriminatory results and simulate physiological conditions. While the NEs provide greater availability of the nonencapsulated CLT form, future studies on the development of the final dosage form of mucoadhesive properties are required to extend vaginal residual time.

## 6. Patents

National and PCT patent applications have been filed for the inventions presented in this publication. Applications have been registered with numbers: P.438501 for UPRP—polish notational patent application and PCT/PL2022/000039 (publication number WO 2023/287309) for PCT patent application.

## Figures and Tables

**Figure 1 pharmaceutics-15-01437-f001:**
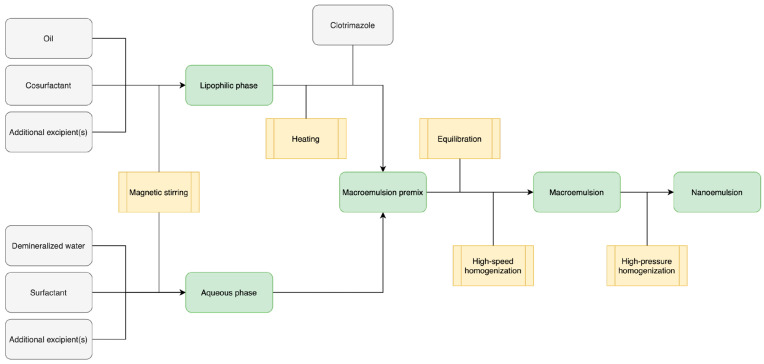
Manufacturing process of selected for further investigation and optimized formulation (S9:1, S8:2, S7:3 and S9:1-CLT, S8:2-CLT, S7:3-CLT).

**Figure 2 pharmaceutics-15-01437-f002:**
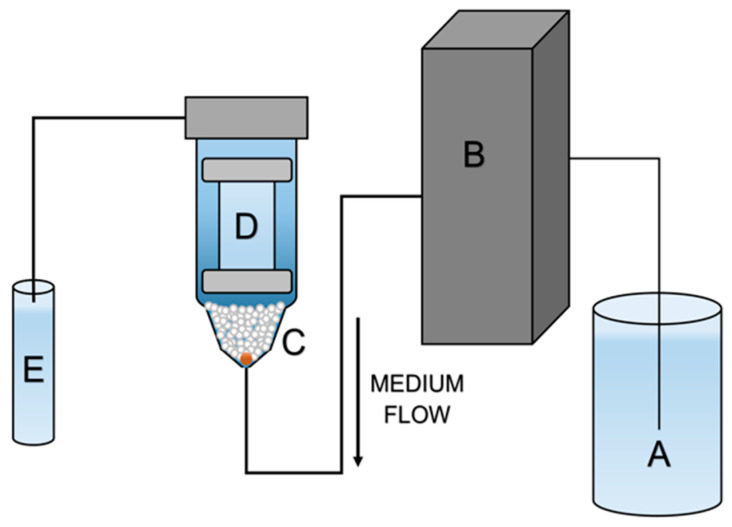
Scheme of flow-through method apparatus. A—medium reservoir, B—piston pump, C—flow-through cells with glass beads, D—dialysis bag with NE inside, E—test tube.

**Figure 3 pharmaceutics-15-01437-f003:**
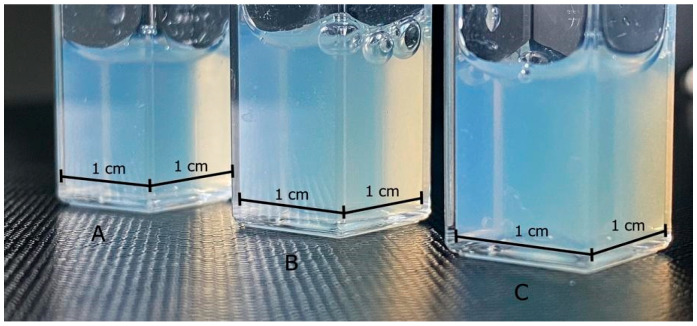
The visual appearance of NEs with CLT. Batches: S9:1-CLT (**A**, left), S8:2-CLT (**B**, centre), S7:3-CLT (**C**, right).

**Figure 4 pharmaceutics-15-01437-f004:**
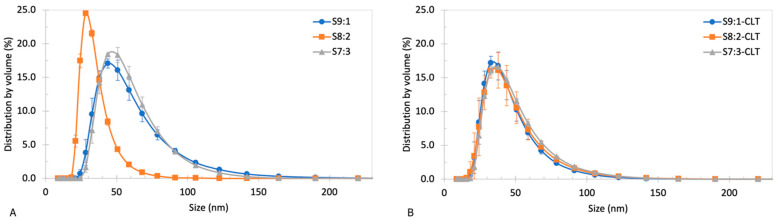
Droplet size distribution by volume of (**A**) blank NEs and (**B**) NEs with CLT. Data expressed as mean values ± standard deviation (MV ± SD, *n* = 3).

**Figure 5 pharmaceutics-15-01437-f005:**
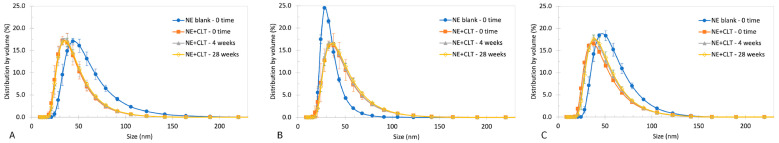
Changes in droplet size distribution by volume for (**A**) S9:1-CLT, (**B**) S8:2-CLT, (**C**) S7:3-CLT. Data expressed as MV ± SD (*n* = 3).

**Figure 6 pharmaceutics-15-01437-f006:**
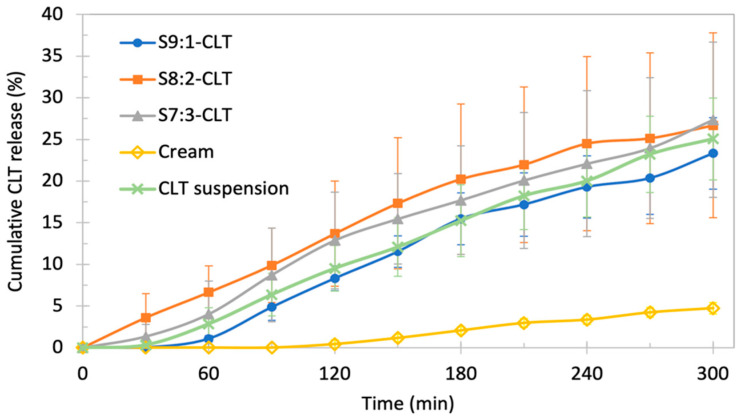
Results of the stationary pilot study—the amount of released API (cumulative) over time. The values are presented as the mean % of released API. Data expressed as MV ± SD (*n* = 3).

**Figure 7 pharmaceutics-15-01437-f007:**
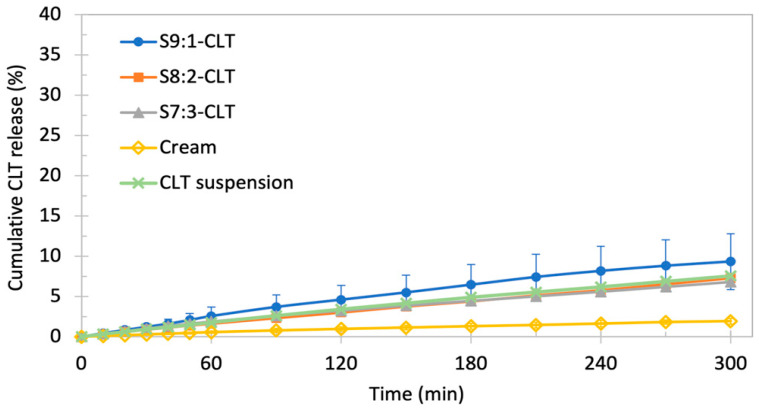
Results of USP IV pilot test—the amount of released API (cumulative) over time. The values are presented as the mean % of released API. Data expressed as MV ± SD (*n* = 3).

**Figure 8 pharmaceutics-15-01437-f008:**
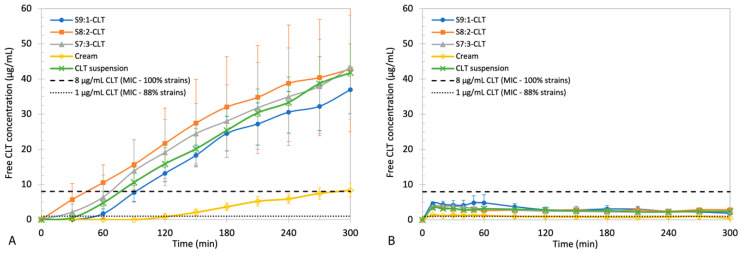
Comparison of the changes in free CLT concentration obtained over time using (**A**) the stationary method and (**B**) the USP IV method with 1 μg/mL CLT MIC (88% isolated strains susceptible) and 8 μg/mL CLT MIC (100% isolated strains susceptible). Data expressed as MV ± SD (*n* = 3).

**Table 1 pharmaceutics-15-01437-t001:** Semiquantitative determination of CLT solubility in tested formulation components.

Solubility Range	<1 mg/g	1–10 mg/g	10–60 mg/g	>60 mg/g
Components	10% Pluronic F-68, 10% Pluronic F-127	Rapeseed oil	Tween 80	PEG 200, Propylene glycol Span 80

**Table 2 pharmaceutics-15-01437-t002:** Compositions and manufacturing methods of the first set of batches.

Number	Oil (% *w*/*w*)	Surfactant(s) (% *w*/*w*)			Additional Excipient(s)		Manufacturing Method(s)	Selection or Rejection ^a^
	Rapeseed	Pluronic F-127	Tween 80	Span 80	PEG 200	Propylene Glycol		
1	40.0	10.0	-	-	-	-	HPH 400	Rejected, Criteria 1
2	40.0	10.0	-	-	-	-	HPH 800	Rejected, Criteria 1
3	40.0	10.0	-	-	-	-	HPH 1200	Rejected, Criteria 1
4	30.0	10.0	-	-	-	-	HPH 800	Rejected, Criteria 1
5	30.0	10.0	-	-	-	-	HPH 1200	Rejected, Criteria 1

^a^—if the formulation was rejected, the rejection criterion is given referring to Section 2.3, i.e., Criteria 1 (translucent/transparent visual appearance) or Criteria 2 (centrifuge stability); HPH—high pressure homogenization; 400—400 ± 100 bar; 800—800 ± 100 bar; 1200—1200 ± 100 bar.

**Table 3 pharmaceutics-15-01437-t003:** Compositions and manufacturing methods of the second set of batches.

Number	Oil (% *w*/*w*)	Surfactant(s) (% *w*/*w*)			Additional Excipient(s)		Manufacturing Method(s)	Selection or Rejection ^a^
	Rapeseed	Pluronic F-127	Tween 80	Span 80	PEG 200	Propylene Glycol		
6	10.0	-	10.0	-	-	-	IM + HSH	Rejected, Criteria 1
7	10.0	10.0	-	-	-	-	IM + HSH	Rejected, Criteria 1

^a^—if the formulation was rejected, the rejection criterion is given referring to Section 2.3, i.e., Criteria 1 (translucent/transparent visual appearance) or Criteria 2 (centrifuge stability); HSH—High Speed Homogenization at 13,000 ± 200 rpm; IM—initial mixing with HSH at low speed—1500 rpm.

**Table 4 pharmaceutics-15-01437-t004:** Compositions and manufacturing methods of the third set of batches.

Number	Oil (% *w*/*w*)	Surfactant(s) (% *w*/*w*)			Additional Excipient(s)		Manufacturing Method(s)	Selection or Rejection ^a^
	Rapeseed	Pluronic F-127	Tween 80	Span 80	PEG 200	Propylene Glycol		
8	40.0	10.0	-	-	20.0	-	HSH + HPH 800	Rejected, Criteria 1
9	40.0	20.0	-	-	10.0	-	HSH + HPH 800	Rejected, Criteria 1

^a^—if the formulation was rejected, the rejection criterion is given referring to Section 2.3, i.e., Criteria 1 (translucent/transparent visual appearance) or Criteria 2 (centrifuge stability); HSH—High Speed Homogenization at 13,000 ± 200 rpm; HPH—high pressure homogenization; 800—800 ± 100 bar.

**Table 5 pharmaceutics-15-01437-t005:** Compositions and manufacturing methods of the fourth set of batches.

Number	Oil (% *w*/*w*)	Surfactant(s) (% *w*/*w*)			Additional Excipient(s)		Manufacturing Method(s)	Selection or Rejection ^a^
	Rapeseed	Pluronic F-127	Tween 80	Span 80	PEG 200	Propylene Glycol		
10	40.0	10.0	10.0	-	-	-	HPH 400	Rejected, Criteria 1
11	30.0	5.0	5.0	-	-	-	HPH 800	Rejected, Criteria 1

^a^—if the formulation was rejected, the rejection criterion is given referring to Section 2.3, i.e., Criteria 1 (translucent/transparent visual appearance) or Criteria 2 (centrifuge stability); HPH—high pressure homogenization; 400—400 ± 100 bar; 800—800 ± 100 bar.

**Table 6 pharmaceutics-15-01437-t006:** Compositions and manufacturing methods of the fifth set of batches.

Number	Oil (% *w*/*w*)	Surfactant(s) (% *w*/*w*)			Additional Excipient(s)		Manufacturing Method(s)	Selection or Rejection ^a^
	Rapeseed	Pluronic F-127	Tween 80	Span 80	PEG 200	Propylene Glycol		
12	20.0	4.5	-	4.5	0.5	-	HSH + HPH 800	Rejected, Criteria 1
13	40.0	5.0	-	5.0	-	0.5	HSH + HPH 800	Rejected, Criteria 1
14	20.0	4.5	-	4.5	-	-	HSH + HPH 800	Rejected, Criteria 1

^a^—if the formulation was rejected, the rejection criterion is given referring to Section 2.3, i.e., Criteria 1 (translucent/transparent visual appearance) or Criteria 2 (centrifuge stability); HSH—High Speed Homogenization at 13,000 ± 200 rpm; HPH—high pressure homogenization; 800—800 ± 100 bar.

**Table 7 pharmaceutics-15-01437-t007:** Compositions and manufacturing methods of the sixth set of batches.

Number	Oil (% *w*/*w*)	Surfactant(s) (% *w*/*w*)			Additional Excipient(s)		Manufacturing Method(s)	Selection or Rejection ^a^
	Rapeseed	Pluronic F-127	Tween 80	Span 80	PEG 200	Propylene Glycol		
15	40.0	18.0	-	2.0	6.6	-	HSH + HPH 800	Rejected, Criteria 1
16	30.0	18.0	-	2.0	6.6	-	HSH + HPH 800	Rejected, Criteria 2

^a^—if the formulation was rejected, the rejection criterion is given referring to Section 2.3, i.e., Criteria 1 (translucent/transparent visual appearance) or Criteria 2 (centrifuge stability); HSH—High Speed Homogenization at 13,000 ± 200 rpm; HPH—high pressure homogenization; 800—800 ± 100 bar; 1200—1200 ± 100 bar.

**Table 8 pharmaceutics-15-01437-t008:** Compositions and manufacturing methods of the seventh set of batches.

Number	Oil (% *w*/*w*)	Surfactant(s) (% *w*/*w*)			Additional Excipient(s)		Manufacturing Method(s)	Selection or Rejection ^a^
	Rapeseed	Pluronic F-127	Tween 80	Span 80	PEG 200	Propylene Glycol		
17	20.0	18.0	-	2.0	6.6	-	HSH + HPH 800	Rejected, gelling
18	20.0	16.0	-	4.0	6.6	-	HSH + HPH 800	Rejected, gelling
19	20.0	14.0	-	6.0	6.6	-	HSH + HPH 800	Rejected, gelling

^a^—if the formulation was rejected, the rejection criterion is given referring to Section 2.3, i.e., Criteria 1 (translucent/transparent visual appearance) or Criteria 2 (centrifuge stability); HSH—High Speed Homogenization at 13,000 ± 200 rpm; HPH—high pressure homogenization; 800—800 ± 100 bar.

**Table 9 pharmaceutics-15-01437-t009:** Compositions and manufacturing methods of the eighth set of batches.

Number	Oil (% *w*/*w*)	Surfactant(s) (% *w*/*w*)			Additional Excipient(s)		Manufacturing Method(s)	Selection or Rejection
	Rapeseed	Pluronic F-68	Tween 80	Span 80	PEG 200	Propylene Glycol		
20	20.0	18.0	-	2.0	6.6	-	HSH + HPH 800	Selected
21	20.0	16.0	-	4.0	6.6	-	HSH + HPH 800	Selected
22	20.0	14.0	-	6.0	6.6	-	HSH + HPH 800	Selected

HSH—High Speed Homogenization at 13,000 ± 200 rpm; HPH—high pressure homogenization; 800—800 ± 100 bar.

**Table 10 pharmaceutics-15-01437-t010:** Compositions of blank NEs.

Ingredients [% *w*/*w*]	Batch		
	S9:1	S8:2	S7:3
Rapeseed oil	20.0	20.0	20.0
Demineralized water	52.8	52.8	52.8
88.9% Lactic acid	0.6	0.6	0.6
Pluronic F-68	18.0	16.0	14.0
Span 80	2.0	4.0	6.0
PEG 200	6.6	6.6	6.6

**Table 11 pharmaceutics-15-01437-t011:** Compositions of NEs with CLT.

Ingredients [% *w*/*w*]	Batch		
	S9:1-CLT	S8:2-CLT	S7:3-CLT
Rapeseed oil	20.0	20.0	20.0
Clotrimazole	1.0	1.0	1.0
Demineralized water	51.8	51.8	51.8
88.9% Lactic acid	0.6	0.6	0.6
Pluronic F-68	18.0	16.0	14.0
Span 80	2.0	4.0	6.0
PEG 200	6.6	6.6	6.6

**Table 12 pharmaceutics-15-01437-t012:** The efficiency of CLT loading. Data expressed as mean value (MV) ± standard deviation (SD) of CLT loading of S9:1-CLT, S8:2-CLT, and S7:3-CLT.

Pressure Level [bar]	Efficiency of CLT Loading [%]
1300 ± 100	97 ± 2
1500 ± 100	53 ± 2
1700 ± 100	47 ± 4

**Table 13 pharmaceutics-15-01437-t013:** Physico-chemical properties of the blank NEs and NEs with CLT (MV ± SD, *n* = 3).

Batch	Z-Average [nm]	PDI	ζ-Potential [mV]	Osmolality [mOsm/kg]	pH
S9:1	67.39 ± 0.32	0.130 ± 0.007	−4.63 ± 0.34	–	3.61
S9:1-CLT	52.16 ± 0.30	0.145 ± 0.010	22.00 ± 1.86	871 ± 25	4.18
S8:2	58.13 ± 0.34	0.147 ± 0.008	16.50 ± 0.25	–	3.55
S8:2-CLT	55.49 ± 0.44	0.159 ± 0.009	19.70 ± 1.50	852 ± 17	4.06
S7:3	64.01 ± 0.11	0.089 ± 0.012	−2.50 ± 0.15	–	3.49
S7:3-CLT	56.11 ± 0.27	0.132 ± 0.005	22.90 ± 0.27	758 ± 5	4.01

**Table 14 pharmaceutics-15-01437-t014:** Physico-chemical properties of NEs with CLT vs. time (MV ± SD, *n* = 3).

Batch	Time [Weeks]	Z-Average [nm]	PDI	ζ-Potential [mV]	pH
S9:1-CLT	0	52.16 ± 0.30	0.145 ± 0.010	22.00 ± 1.86	4.18
	4	52.80 ± 0.06	0.141 ± 0.008	17.40 ± 0.10	4.17
	28	53.22 ± 0.17	0.143 ± 0.014	21.60 ± 1.49	4.13
S8:2-CLT	0	55.49 ± 0.44	0.159 ± 0.009	19.70 ± 1.50	4.06
	4	54.62 ± 0.27	0.140 ± 0.010	18.40 ± 0.79	4.06
	28	55.61 ± 0.15	0.131 ± 0.006	24.20 ± 0.97	4.07
S7:3-CLT	0	56.11 ± 0.27	0.132 ± 0.005	22.90 ± 0.27	4.01
	4	56.20 ± 0.20	0.119 ± 0.004	22.20 ± 0.81	4.02
	28	56.77 ± 0.31	0.116 ± 0.006	23.50 ± 2.46	3.93

## Data Availability

Data is contained within the article.

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
