# Peer review of "Nanoemulsion Loaded with Clotrimazole Based on Rapeseed Oil for Potential Vaginal Application—Development, Initial Assessment, and Pilot Release Studies"

_pharmaceutics, 2023, doi:10.3390/pharmaceutics15051437_

Round 1

Reviewer 1 Report (New Reviewer)

Manuscript entitled “Nanoemulsion loaded with clotrimazole based on rapeseed oil for potential vaginal application – development, initial assessment and pilot release studies” is, in my opinion, a well-conceived work with logically chosen methods. The discussion follows logically from the presented results. It is obvious that the manuscript underwent significant changes, especially the part related to the way of presenting the results, which improved it. The authors made an effort to make all the corrections in the appropriate places and clearly show all the changes made. I believe that the work can be published in the presented form considering the results obtained and the conclusions derived from them.

Author Response

Dear Reviewer!

Thank you for your comments on our manuscript. The point-by-point answer for your opinion on the previous version of the manuscript is presented below:

Comment 1:

Manuscript entitled “Nanoemulsion loaded with clotrimazole based on rapeseed oil for potential vaginal application – development, initial assessment and pilot release studies” is, in my opinion, a well-conceived work with logically chosen methods. The discussion follows logically from the presented results. It is obvious that the manuscript underwent significant changes, especially the part related to the way of presenting the results, which improved it. The authors made an effort to make all the corrections in the appropriate places and clearly show all the changes made. I believe that the work can be published in the presented form considering the results obtained and the conclusions derived from them.

Answer 1: We really appreciate the opinion of Reviewer on our manuscript. We are grateful for the time the Reviewer spent on reading our work.

Reviewer 2 Report (New Reviewer)

The manuscript submitted by Smoleński et al. reports on the optimization of nanoemulsion formulation with clotrimazole destined for vaginal application. The topic fits under the scope of the Journal and is interesting particularly for the scientific community in pharmaceutical technology and pharmaceutical industry. However, there are some comments that should be addressed to strengthen the manuscript and increase its clarity.

Since there are no lines inserted in the manuscript, it is difficult to strictly address the comment, particularly because the manuscript is revised  (track changes in the text).

Introduction:

please keep uniformity in the style; Candida spp., Candida is in italic or not? Check throughout the whole manuscript and re-write where required.

“Bachhav and Patravale as well as Soriano-Ruiz et al.”-please indicate ref. number just after indicating the authors, not at the end of the sentence.

Indicate the physicochemical properties of clotrimazole that are important for selection of ingredients of formulation and during its development (solubility, Mw).  It would also be beneficial to denote possible permeability data for clotrimazole (these data are relevant considering further development of formulation destined for local vaginal therapy).

Title of subsection 2.5. is too long, try to reduce it.

Results and discussion:

To improve the clarity, in my opinion, it would be worthy to prepare a table indicating composition of different batches evaluated during screening and optimization of the nanoemulsion formulation. This table should be placed in the Experimental part. Then, it would be easier to follow a discussion of the results obtained. The number of tables in Results will be then reduced and the only the main results discussed.

Page 18: When discussing the optimal size regarding penetration into mucosa (mucus), could you  please add what did you mean by larger particles (which size indicate larger particles in that context)? You should be careful with the size nanoemulsion/nanoparticles, because of the problem of mucus desquamation, and smaller size nanoncarriers could be lost with the mucus.

How did you detect efficiency of clotrimazole loading (Table 12)? I cannot find the procedure in the experimental part. Please indicate.

The formulation is intended for vaginal administration. What about viscosity of the nanoemulsion, is it adequate for vaginal application? Could you provide more discussion on that issue?

Author Response

Dear Reviewer!

Thank you for your interesting comments and suggestions. The point-by-point answer for your opinion on the previous version of the manuscript is presented below:

Comment 1:

The manuscript submitted by Smoleński et al. reports on the optimization of nanoemulsion formulation with clotrimazole destined for vaginal application. The topic fits under the scope of the Journal and is interesting particularly for the scientific community in pharmaceutical technology and pharmaceutical industry. However, there are some comments that should be addressed to strengthen the manuscript and increase its clarity.

Since there are no lines inserted in the manuscript, it is difficult to strictly address the comment, particularly because the manuscript is revised (track changes in the text).

Answer 1: We added numbers to the lines. The manuscript was uploaded in the track changes mode as we were instructed to do so in resubmission process.

Comment 2:

Introduction:

please keep uniformity in the style; Candida spp., Candida is in italic or not? Check throughout the whole manuscript and re-write where required.

Answer 2: All of the Candida spp. style were unified.

Comment 3:

“Bachhav and Patravale as well as Soriano-Ruiz et al.”-please indicate ref. number just after indicating the authors, not at the end of the sentence.

Answer 3:  The sentence has been changed as follows - Bachhav and Patravale [26] as well as Soriano-Ruiz et al. [22,27] have developed microemulsion-based gel, nanoemulsion (NE) and multiple-emulsion with CLT.

Comment 4:

Indicate the physicochemical properties of clotrimazole that are important for selection of ingredients of formulation and during its development (solubility, Mw).  It would also be beneficial to denote possible permeability data for clotrimazole (these data are relevant considering further development of formulation destined for local vaginal therapy).

Answer 4: The physicochemical properties of CLT has been included in the manuscript as follows - CLT is a weak base (pKa – 4.7 and 6.0) with a molecular weight of 344.8 g/mol [14]. Despite its high antifungal activity, the possibilities of its applications are limited due to poor water solubility (0.49 mg/L) and lipophilic properties (log P – 6.1) [14,15].

Comment 5:

Title of subsection 2.5. is too long, try to reduce it.

Answer 5:

The title of subsection 2.5 has been changed into Manufacturing of experimental S9:1, S8:2 and S7:3 blank formulations and corresponding batches with CLT for further assessment.

Comment 6:

Results and discussion:

To improve the clarity, in my opinion, it would be worthy to prepare a table indicating composition of different batches evaluated during screening and optimization of the nanoemulsion formulation. This table should be placed in the Experimental part. Then, it would be easier to follow a discussion of the results obtained. The number of tables in Results will be then reduced and the only the main results discussed.

Answer 6:

We very much appreciate the suggestion. We agree that one table would increase the clarity, however, this would also compromise the scientific outcome of the manuscript, as one of the main aims of the study was to present the process of formulation development so other researches could understand the process and develop their nanoemulsions. Placing the table in Experimental section would lead to separation of compositions from their performance, which in this case is necessary to visualize the process of development.  Therefore, we decided to keep the current structure.

Comment 7:

Page 18: When discussing the optimal size regarding penetration into mucosa (mucus), could you  please add what did you mean by larger particles (which size indicate larger particles in that context)? You should be careful with the size nanoemulsion/nanoparticles, because of the problem of mucus desquamation, and smaller size nanoncarriers could be lost with the mucus.

Answer 7: This is very interesting comment. By larger particles we meant those of 200-500 nm sized. CLT is intended to act locally, its absorption into circulatory system is not beneficial. To reduce the risk of side effect and increases intramucosal residual time of nanodroplets we decided to aim into < 100 nm diameter of oil droplets. It would be very interesting to design the clinical study evaluating the efficiency of treatment depending on the size of nanodroplets.

The sentence was clarified as follows - Larger droplets (200-500 nm) would penetrate to vaginal mucosa more rapidly and hence increase absorption of the CLT to the circulatory system, which in this case is not considered beneficial due to the possible systemic side effects [17].

Comment 8:

How did you detect efficiency of clotrimazole loading (Table 12)? I cannot find the procedure in the experimental part. Please indicate.

Answer 8: The efficiency of clotrimazole loading presented in table 12 was assessed using method described in subsection 2.5., the samples were processed using the assay method described in subsection 2.9 and analyzed using HPLC method (subsection 2.10).

The text has been updated for better clarity as follows - The results of the drug loading efficiency depending on different levels of pressure during HPH (following the procedure 2.5 and 2.9) are presented in Table 12.

Comment 9:

The formulation is intended for vaginal administration. What about viscosity of the nanoemulsion, is it adequate for vaginal application? Could you provide more discussion on that issue?

Answer 9: The obtained nanoemulsion was of low viscosity. The scope of this paper was to develop initial form of the vaginal formulation. Emulsion-based vaginal dosage forms are usually developed into emulgels or pessaries :

  • https://doi.org/10.1208/s12249-016-0652-6 (ref. 36),
  • https://doi.org/10.15171/apb.2017.073 (ref. 42),
  • https://doi.org/10.1016/j.colsurfb.2012.10.038 (ref. 43),
  • https://innovareacademics.in/journals/index.php/ijpps/article/view/9885 (ref. 26).

The development of the final form is a scope of another study where we would evaluate the applicability, properties and performance of different selected dosage forms based on obtained nanoemulsion described in this paper. Including these aspects in this paper would result in limitation of currently presented data as the manuscript would be too long if both parts were included.  In this paper we presented and discussed the technological aspect of nanoemulsion manufacturing followed by the assessment of initial product, hence, the important aspect of viscosity brought by the Reviewer has not been assessed.

Reviewer 3 Report (New Reviewer)

In the authors' work, nanoemulsions of rapeseed oil were designed and evaluated for characterization and in vitro release. The current revised version has significant improvements and is recommended to be published after the following issues have been addressed.

1. There are some textual presentation issues, e.g. symbols characterising the range of values should not use the short hyphen, spaces are missing between some values and units, and some abbreviations should be labelled with their full names (e.g. API, ACN) when first used, etc. Please check the full text and make corrections.

2. When conducting in vitro release studies, attention needs to be paid to whether the release medium meets the leaky tank conditions.

3. The number of samples (n = ? ) used for calculation should be added after the title of figures and tables.

4. As only in vitro studies are available and the irritation issues focused on mucosal administration are uncertain, it is recommended that the biosafety of the components of the formulation be discussed in detail.

5. Fewer studies have been reported on vaginal routes of administration than injections, oral and transdermal administration. In a previous report (Int J Pharm. 2020 Aug 30;586:119616. doi: 10.1016/j.ijpharm.2020.119616.), two nanoparticles dispersed in a temperature-sensitive gel for vaginal administration were used to achieve a reduced toxicity and increased efficacy purpose and is proposed to be cited in the next revision.

Minor editing of English language required.

Author Response

Dear Reviewer!

Thank you for your interesting comments and suggestions. The point-by-point answer for your opinion on the previous version of the manuscript is presented below:

General Comment:

In the authors' work, nanoemulsions of rapeseed oil were designed and evaluated for characterization and in vitro release. The current revised version has significant improvements and is recommended to be published after the following issues have been addressed.

Answer: We really appreciated all of the comments and suggestions brought by Reviewer. We believe they will help us improve our work.

Comment 1:

There are some textual presentation issues, e.g. symbols characterising the range of values should not use the short hyphen, spaces are missing between some values and units, and some abbreviations should be labelled with their full names (e.g. API, ACN) when first used, etc. Please check the full text and make corrections.

Answer 1: The manuscript was checked, and all necessary corrections were made.

Comment 2:

When conducting in vitro release studies, attention needs to be paid to whether the release medium meets the leaky tank conditions.

Answer 2: To summarize the information from the manuscript - the medium was chosen to provide both sufficient clotrimazole solubility and simulation of physiological pH. To provide sufficient solubility of CLT. The volume of PBS pH 3.5 was set to 300 mL. The nanoemulsion was placed in the dialysis bag sealed with clips. One of the clips had built in metal bar which positioned the bag vertically. The bags were checked for macro leaks; however, the micro leaking could not be noticed. The samples with milky medium were excluded. The dialysis membrane was chosen based on a low CLT sorption to its material.

Comment 3:

The number of samples (n = ? ) used for calculation should be added after the title of figures and tables.

Answer 3: All corrections have been done. The information was added to relevant tables and figures.

Comment 4:

As only in vitro studies are available and the irritation issues focused on mucosal administration are uncertain, it is recommended that the biosafety of the components of the formulation be discussed in detail.

Answer 4: The extended discussion of components was included. The manuscript has been updated as follows - Despite the osmolality of NEs fulfilling the recommendations of WHO, the potential irritating properties of components was investigated. Surfactants in high concentrations are recognized as potentially irritating or even toxic substances [67]. Implementation of Pluronic F-68 in high content (> 20%) in vaginal formulation containing nanoparticles was previously investigated by Zhang et al. [68], where no irritating properties of this excipient was observed in vivo on rabbits. Safety of Span-80 was evaluated by Minamisakamoto et al. [69] in the study of noisome, where insignificant toxicity of high (46.5%) concentration of the surfactant was observed. This indicates NEs should not exhibit irritating properties on vaginal mucosa as concentration of surfactants used in our study are lower than reported in previous research.

Comment 5:

  1. Fewer studies have been reported on vaginal routes of administration than injections, oral and transdermal administration. In a previous report (Int J Pharm. 2020 Aug 30;586:119616. doi: 10.1016/j.ijpharm.2020.119616.), two nanoparticles dispersed in a temperature-sensitive gel for vaginal administration were used to achieve a reduced toxicity and increased efficacy purpose and is proposed to be cited in the next revision.

Answer 5: The article was included in the discussion as follows - Implementation of Pluronic F-68 in high content (> 20%) in vaginal formulation containing nanoparticles was previously investigated by Zhang et al. [68], where no irritating properties of this excipient was observed in vivo on rabbits.

Reviewer 4 Report (New Reviewer)

The manuscript (pharmaceutics-2342975) entitled "Nanoemulsion loaded with clotrimazole based on rapeseed oil for potential vaginal application – development, initial assessment and pilot release studies" is interesting but the following suggestions would be helpful to further improve:

1. Key influencing factors responsible to affect biopharmaceutical performance of generated should be highlighted and clearly represented for the quick understanding of the reader. 

2. Factors influencing the release performance of generated nanoemulsion should also be highlighted and clearly represented for the quick understanding of the reader.

It is ok, typo/grammatical errors should be rectified sincerely.

Author Response

Dear Reviewer!

Thank you for your interesting comments and suggestions. The point-by-point answer for your opinion on the previous version of the manuscript is presented below:

General comment :

The manuscript (pharmaceutics-2342975) entitled "Nanoemulsion loaded with clotrimazole based on rapeseed oil for potential vaginal application – development, initial assessment and pilot release studies" is interesting but the following suggestions would be helpful to further improve:

Answer : We really appreciated all of the comments and suggestions brought by Reviewer. We believe they will help us improve our work.

Comment 1:

  1. Key influencing factors responsible to affect biopharmaceutical performance of generated should be highlighted and clearly represented for the quick understanding of the reader.

Answer 1: The part in introduction was revised to underline key factors - The key factors affecting biopharmaceutical performance are mean size of droplets (z-average), their uniformity, physiological pH and osmolality, therefore  the optimal NE should have: a) the droplet size < 90 nm to increase gravitational stability (phase separation due to coalescence and flocculation), b) the absolute value of ζ-potential should be 20 or higher to prevent droplets aggregation, c) polydispersity index < 0.3 to provide sufficient uniformity, d) physiological pH from 3.5 to 4.5, and e) osmolality < 1200 mOsm/kg [47–50].

And in discussion part the summary has been added as follows - To summarize, obtained NEs have met the initial requirements resulting from key factors of biopharmaceutical performance i.e., z-average of 50–60 nm, narrow size-distribution (high uniformity of the droplet size), physiological pH of approx. 4.0 and osmolality within the range of 700–900 mOsm/kg.

Comment 2:

  1. Factors influencing the release performance of generated nanoemulsion should also be highlighted and clearly represented for the quick understanding of the reader.

Answer 2: The phrase has been added to highlight the key factors - The most important factors characterizing NEs performance within the study of the release of free CLT from encapsulated form are the fast onset of CLT release from NE and the ability to reach MIC or higher CLT concentration.

Further investigation of the factors is not possible as both dissolution tests were not discriminatory. The next range of tests to develop discriminatory and more biorelevant method is planned.

Comment 3:

Comments on the Quality of English Language

It is ok, typo/grammatical errors should be rectified sincerely.

Answer 3: A manuscript proofreading was performed and typo/grammatical errors have been corrected.

This manuscript is a resubmission of an earlier submission. The following is a list of the peer review reports and author responses from that submission.

Round 1

Reviewer 1 Report

The manuscript entitled “Nanoemulsion loaded with clotrimazole based on vegetable oil 2 for potential vaginal application – development, initial assessment and pilot release studies” describes the development of clotrimazole nanoparticle formulations using a high-speed and high-pressure homogenizer. Although the manuscript is well written, but still needs some improvement on the following points:

  1. How the final formulations (nano-emulsion) were designed for the vaginal route of admiration as it is in liquid of low viscosity (short contact time) which makes the formulation not visible for vaginal administration.
  2. In the stability study, humidity range in the storage conditions is very wide (30-65%). Also, what type of container did you use to store your formulation, and did the humidity influence the stability of your formulations in this container?
  3. In table 4, is the impact of the high pressure applicable for all three formulations, as the results in this table do not specify the drug content of which formulation.

Reviewer 2 Report

The manuscript reports the study of development of vaginal formulations, followed by the characterization.

The thesis is well organized but I feel that some parts need to be revised before it can be published.

The major criticism is represented by the fact that a systematic study of the formulation does not appear: the authors declare that they have set up 73 formulations and have characterized only 3 of them as demonstrated to be the most stable according to two established criteria.

It would certainly be useful to have an overview of the prepared formulations (also summarized in a diagram) to understand what the study was based on. For example, a design of experiments (DOE) or a ternary diagram could be made. Was it only the ratio between surfactant and cosurfactant or also between surfactant and oil that changed? And what effect does PEG have on stability? have formulations been prepared without it or with different concentrations?

I believe that all these tests have been done by researchers and therefore should be shown.

Another aspect that needs to be better described is that relating to the determination of the loading efficiency as a function of the pressure in HPH method. What formulation were they made on? or are these loading efficiencies at the relative pressures constant for all 73 formulations?

The authors also conclude that the release studies should be investigated as the medium used is much more abundant than the amount of fluid physiologically present in the vagina. In this regard, they have faced two types of assay, but in one case they have used a large dissolution volume (perhaps it could have been smaller) and in the other they have used a flow of medium considerably higher than the physiological turnover (about 6 ml per day as they themselves report) all justified as a property of the peristaltic pump used. But also in other parts of the discussion the authors report that the results should be further investigated. I believe that some factors of the experiments are critical and these lead to results that need to be deepened making their discussion factitious.

Other minor points can be noted:

- line 245: what vehicle was used to prepare the suspension?

- line 256-257: what do you think could be the cause of the deterioration of the regenerated cellulose membrane? these are generally quite compatible also with solvents. In any case, authors can try to replace it with another type so as not to be forced to discard formulations.

- Tab. 5: could the authors explain which components, in their opinion, could contribute to giving those osmolarity values? Has it been taken into consideration that the instrument may not be suitable for measurements in emulsions?

- Tab. 5: the cause of the different values of zeta potential obtained for the blank NEs is not analyzed in the discussion. The difference in the composition is minimal while there are notable differences which, moreover, do not even reflect the composition. If an ingredient stabilizes or destabilizes NE the measured value of zeta potential should be in accordance with the concentration of its presence.

- Figures: indicate what the error bars mean

- line 401: if the standard deviations are high perhaps there would be a need to increase the number of tests.

- line 488: also for the formulation S8:2 smaller diameters are measured, but to a lesser extent

Reviewer 3 Report

The manuscript is of great interest for the manufacturing of nanoemulsions and the comparision between drug release testing methods.

However, it lacks of interest for a clinical application of the formulations developed.

My main concerns about the manuscript are:

1. Why do you affirm that batch S8:2-CLT is the most suitable one? as observed by the data, there are no significant differences between the three formulations developed. The selection of batch S8:2-CLT must be supported.

2. The droplet size distribution of blank S8:2 is clearly different to the other formulations presented in the manuscript (it is clearly observed in figures 4 and 6). Which is the reason for such a great difference? it has been confirmed in several batches to affirm that it is not a wrong measurement? And then, how the authors explain this difference?

3. The last and most important concern is about the clinical application. You refer to CLT suspension as not suitable for vaginal administration due to poor vaginal retention. That is why CLT cream is used. Your nanoemulsions proved to provide a drug release similar to vaginal suspensions, so the improve the slow drug release from creams. But, does your formulations improve vaginal retention compared to suspensions? This must be evaluated. A bioadhesion test would be the most suitable way, but if it is not possible, at least viscosity and consistency measurements should be provided evaluate if these parameters are more similar to suspensions or to creams. Otherwise, your formulation does not provide any clinical improvement.

Finally, I have found some typos and format errors that authors should revise:

- In the introduction, line 39, "RCV" should be changed by "RVC"

- In the introduction, line 69, in vitro appears with just "in" in italics. Revise the use of italics each time "in vitro" is used throughout the text.

- In materials and methods, the format of subsection 2.2. heading is not the same as the other subsections.

- In materials and methods, line 176, you do not include the variation for the last pressure condition (1700 +- ¿?) 

Round 2

Reviewer 1 Report

The authors answered all of my questions adequately.

Thanks,

Reviewer 2 Report

The authors have attempted to address the major criticisms made but I do not think they have improved the manuscript; issues remain the same.

The table added in the supplementary material is still not useful as it does not reflect a systematic study for the development of the formulation.

Many assays have been performed but they are not definitive, they are only preliminary or screening and need further investigation. This does not allow the authors to make an adequate and detailed discussion of their discoveries, diminishes the work and makes it less scientifically of little use.

Therefore, if the reader has no information on how to develop the best formulation but only that, among the many preparations, some are more stable than others; if the technological characterization is still preliminary and needs to be further explored and if the release essays are to be repeated, I ask the authors: what useful information can we draw from this manuscript?

Among other things, I find the explanation given by the authors to justify the high standard deviation values obtained in the release tests rather specious. The molecular cut off of the membrane is 10,000 daltons, how can this affect the passage of a drug such as CLT? this variability is instead not evident in the release from the cream and less marked in that from the suspension.

Reviewer 3 Report

The manuscript has been properly revised